# Sulfated Galactans from Agarophytes: Review of Extraction Methods, Structural Features, and Biological Activities

**DOI:** 10.3390/biom13121745

**Published:** 2023-12-05

**Authors:** Khosook Chumsook, Jantana Praiboon, Xiaoting Fu

**Affiliations:** 1State Key Laboratory of Marine Food Processing & Safety Control, College of Food Science and Engineering, Ocean University of China, Qingdao 266003, China; khosook.ch@ku.th; 2Department of Fishery Science and Technology (International) Program, Kasetsart University, Bangkok 10900, Thailand; 3Department of Fishery Biology, Kasetsart University, Bangkok 10900, Thailand; ffisjtn@ku.ac.th

**Keywords:** sulfated galactans, extraction, structure, bioactivity, agarophytes

## Abstract

Agarophytes are important seaweeds of the Rhodophyta type, which have been highly exploited for industrial use as sources of a widely consumed polysaccharide of agar. In addition to that, sulfated galactans (SGs) from agarophytes, which consist of various functional sulfate groups, have attracted the attention of scientists in current studies. SGs possess various biological activities, such as anti-tumor, anticoagulant, anti-inflammatory, antioxidant, anti-obesity, anti-diabetic, anti-microbial, anti-diarrhea, and gut microbiota regulation properties. Meanwhile, the taxonomy, ecological factors, i.e., environmental factors, and harvest period, as well as preparation methods, i.e., the pretreatment, extraction, and purification conditions, have been found to influence the chemical compositions and fine structures of SGs, which have, further, been shown to have an impact on their biological activities. However, the gaps in the knowledge of the properties of SGs due to the above complex factors have hindered their industrial application. The aim of this paper is to collect and systematically review the scientific evidence about SGs and, thus, to pave the way for broader and otherwise valuable industrial applications of agarophytes for human enterprise. In the future, this harvested biomass could be sustainably used not only as a source of agar production but also as natural materials in functional food and pharmaceutical industries.

## 1. Introduction

Red seaweeds used for agar production are called “agarophytes” due to their specific cell wall polysaccharides, which are water-soluble dietary fibers and nutraceuticals [1]. The main genera of agar sources belong to *Gelidium*, *Gracilaria*, *Porphyra*, and *Ahnfeltia*. *Gracilaria* and *Gelidium* are the principal seaweeds used for the industrial production of agar according to the report of the seaweed hydrocolloid market [2]. *Gracilaria* is one of the most cultivated biomasses in the world and is the preferred seaweed for food-grade agar production [3]. According to the FAO data, it ranks third in the global aquaculture production of farmed seaweed [4]. Though *Gelidium* is a wild seaweed with limited natural stocks, it is still the preferred seaweed for making high-quality bacteriological-grade agar and agarose [5]. Porphyra and *Ahnfeltia* are used at significantly smaller industrial scales compared to those of *Gracilaria* and *Gelidium*. Agar is sometimes produced from *Porphyra* in China by using *P. haitanensis* from the late growing period [6,7,8] and in Japan by applying discolored *P. yezoensis* [9,10], both of which are low-commercial-value materials for *Porphyra* food production. In Russia, *Ahnfeltia* species of *A. tobuchiensis* and *A. plicata* are used in the domestic agar industry [11]. In addition to the production of agar, agarophytes are notable for their potential applications in the functional food and pharmaceutical industries [12,13,14]. Sulfated galactans (SGs) from agarophytes are polysaccharides which are composed of 3-linked β-d-galactopyranosyl and 4-linked α-l-galactopyranoyl or 3,6-linked anhydro-α-l-galactopyranosyl, harboring functional sulfate, methyl, and/or pyruvate groups in their saccharide units [15,16]. Sulfated galactan from *Porphyra* is termed Porphyran [17,18]. With the trend of the utilization of seaweed polysaccharides for health, the study of SGs continues to increase, focusing on biochemistry, molecular biology, chemistry, pharmacology, pharmacy, and other related fields (Figure 1) [14].

Various potential biological activities of SGs have been reported, such as anti-tumor [19,20,21,22], anticoagulant [15,23,24], anti-inflammatory [25,26,27,28], antioxidant [29,30,31], lipid metabolism [32,33,34], anti-obesity [35,36,37,38], anti-diabetic, refs. [17,39,40,41] anti-microbial, refs. [42,43,44], immune modulation [45,46,47], gut microbiota regulation [48,49], and anti-diarrhea activities [50,51,52,53,54]. Their biological activities depend on various factors, including the molecular mass, composition of monosaccharides, degree of sulfate esterification, position of sulfate esterification, and degree of polymerization, which are affected by the change in taxonomical positions, morphology, geographical conditions (salinity, temperature, light, and nutrients), the biotic properties (e.g., whether they are epiphytes) of each seaweed genus, and also preparation methods [55,56]. The suitable conditions for the extraction and purification of SGs to obtain high purities, increased bioactivities, and even to discover new bioactivities have been widely studied [57,58].

Jiao et al. [59] reviewed the current progress in the research of sulfated polysaccharides related to their structure diversity, bioactivities, and mechanisms of action. Patel [60] reviewed the therapeutic importance of sulfated polysaccharides of fucan from brown seaweed, carrageenan, and agar from red seaweed as well as ulvan from green seaweed. These reviews comprise broad information on the structure and bioactivities of marine sulfated polysaccharides but give few details about extraction methods. However, there is no comprehensive review of SGs from agarophytes. In this review, scientific evidence of the agarophytes of *Gelidium*, *Gracilaria*, *Porphyra*, and *Ahnfeltia* that covered their purification, structure, and biological activities was summarized, thus providing a reference for the utilization of these agarophytes in broader industrial fields.

## 2. Pretreatment and Extraction of SGs

The sulfation pattern is the major structural variation of SGs, for which the number and position of sulfated groups are quite heterogeneous and specific to agarophytes species, and thus the preparation methods are different [61]. The procedure for the preparation of SGs from agarophytes is summarized in Figure 2, referring to steps and specific methods of pretreatment, extraction, and purification procedures.

### 2.1. Pretreatment

#### 2.1.1. Delipidation

Agarophytes are collected, cleaned, and dried for further use. Besides polysaccharides, agarophytes consist of many biological molecules of proteins, lipids, pigments, etc., affecting the purity and bioactivity of SGs. Wongprasert et al. [45] removed lipids of *Gracilaria fisheri* with benzene and acetone. Wu et al. [62] extracted lipids of the dried *Gracilaria lemaneiformis* by using organic solvents of light petroleum, acetone, and methanol. Xie et al. [63] pretreated six species of red seaweeds by soaking the dried materials in a solution of methanol/dichloromethane/water (4:2:1; *v*/*v*/*v*) by a ratio of 10:1 (*v*/*w*, mL/g) with shaking for 24 h. Supercritical extraction was also adopted to remove lipids from agarophytes. Yu et al. [64] pretreated *Pyropia yezoensis* powder with the supercritical CO_2_ under 45 °C/30 MPa for extraction and 35 °C/5 MPa for separation of lipids with the CO_2_ flow rate of 10 L h^−1^.

#### 2.1.2. Depigmentation

Pigments of agarophytes and their oxidized compounds formed during hot water extraction affect the chromatographic analysis and impede the accurate identification of polysaccharides [65,66]. On the other side, the pigments and phenols are biological compounds, which indicate the necessity of their extraction. *Gracilaria rubra* powder was pretreated with 85% ethanol three times for 3 days to remove pigments and polyphenols [30]. Olasehinde et al. [67] pretreated *Gracilaria gracilis* by soaking in the extraction solvent (ethanol/water and 1 N HCl (1:1, *v*/*v*)) in the ratio 1:20 (*w*/*v*) for 24 h and thereafter filtered the solvent to remove pigments. Small pieces of dried *Pyropia yezoensis* were suspended in 85% ethanol (1:40, *w*/*v*), (Wako Pure Chemical Industries (Tokyo, Japan)), heated at 75 °C for 1 h, filtrated, and the residues were extracted again according to the above procedures to obtain decolorized seaweed [32,68]. Wongprasert et al. [45] performed pretreatment of *G. fisheri* to eliminate pigments by the Soxhlet method using benzene and acetone as the solvent.

Yuan et al. [69] performed the depigmentation of *G. lemaneiformis* by 3-times extraction with 70% ethanol (50:1, *v*/*w*) for 30 min each time. The dried *Gracilaria debilis* was ground and extracted with a mixed solvent of acetone/chloroform/methanol (2:1:1; *v*/*v*/*v*) in a Soxhlet apparatus for 48 h to produce depigmented seaweed [24]. In order to improve the purity of SGs, Khan et al. [70] used the extraction solvent of methanol/dichloromethane/water in the ratio of 4:2:1 (*v*/*v*/*v*) at room temperature for 24 h to pretreat the seaweed material of *Gracilaria chouae*. The above reports indicated that the depigmentation process was able to improve the quality of SGs. On the other hand, the extracts containing lots of natural pigments and phenols showed excellent antioxidant activities, attributing their applications in functional food and cosmetics.

#### 2.1.3. Deproteinization

The elimination of proteins is a key step in the extraction and purification of SGs. The sulfated polysaccharide of *Gracilaria verrucosa* was extracted with water, and the supernatant was mixed with the Sevag reagent (chloroform/n-butanol 4:1, *v*/*v*), the chemicals were purchased from Sigma-Aldrich (St. Louis, MO, USA), in a ratio of 1:5 (*w*/*v*), thoroughly shaken for 30 min, and centrifuged to remove denatured proteins. The deproteinization treatment was repeated five times to remove proteins that might interfere with biological assays [52]. In a similar way, a freeze-dried crude fraction of *Gracilaria corticata* was deproteinized by the Sevag method with chloroform/n-butanol of 5:1 (*v*/*v*) [71]. Sudharsan et al. [24] also reported that after the deproteinization treatment of *G. debilis* by n-butanol/ chloroform (4:1, *v*/*v*), no protein was detected by the Lowry method. The above reports indicated that though the Sevag method was complicated and time-consuming, it was the primarily adopted deproteinization method [72,73]. On the other hand, protein impurities in *Ahnfeltia tobuchiensis* (Kanno et Matsubara) Mak could be removed by 0.5–1.0% Ca(OH)_2_ suspension at 100–120 °C, and the treatment by a higher concentration of Ca(OH)_2_ suspension resulted in a lower amount of protein impurities determining by HPLC [74]. 

### 2.2. Traditional Water Extraction

#### 2.2.1. Hot Water Extraction

Hot water extraction is the most used method for the extraction of SGs. Isaka et al. [25] compared the SGs of two types of *Pyropia yezoensis*, i.e., normal nori (N-P) and discolored waste nori (DC-P), which both sample are provided by Urashima Co. (Kumamoto, Japan) and Japan Fisheries Cooperatives (Saga, Japan), respectively. Dried powders were extracted from the hot water at 95 °C for 1.5 h, followed by precipitation with ethanol. SGs of DC-P and N-P were obtained with a yield of 20.6% (*w*/*w*) and 10.6% (*w*/*w*) and molecular mass of 30 kDa and 224 kDa, respectively, while they showed similar results of sulfate content of 10.5–10.7% (*w*/*w*). Dong et al. [75] studied the effects of main extraction factors on the yield, i.e., the temperature, the time, and the ratio of the raw material to water. The optimal conditions were a temperature of 80 °C, an extraction time of 3 h, and a raw material-to-water ratio of 1:25 (*w*/*v*), which resulted in the highest yield of 20.48%. SGs from *Gracilaria blodgettii* were extracted from the distilled water at 90 °C for 2 h with a yield of 24.12%, a sulfated content of 9.16%, and a molecular weight of 6.149 kDa [70]. Li et al. [34] used citric acid for extraction of SGs from *G. lemaneiformis* at pH 2.3, 100 °C for 2 h with the ratio of 1:25 (*w*/*v*). The extracted SGs showed an Mw of 31.2 kDa, a sulfate content of 25%, and a 3,6-AG content of 15.3%, which exhibited a lipid metabolism-regulating activity in high-fat-diet mice by decreasing lipid levels.

#### 2.2.2. Cold Water Extraction

Besides hot water extraction, the cold water extraction method was also adopted for SG preparation. The SGs isolated from *Gracilaria edulis* by the cold water extraction method showed anticancer activities [76]. In another study, Imjongjairak et al. [77] determined the effect of the temperature on the extraction yield SGS from *G. fisheri*. Crude SGs which were obtained at a high temperature of 55 °C expressed higher values of the yield and the sulfate content of 9.66% and 8.95% than those obtained at a room temperature of 25 °C with the yield and the sulfate content of 5.41% and 6.86%, respectively. Moreover, the anti-oxidative activity of crude SGs extracted from room temperature was higher than that extracted at a higher temperature. Wongprasert et al. [45] extracted SGs from *G. fisheri* at 35–40 °C for 4 h with a yield of 3% with a sulfate content of 12.7% and a molecular weight of 100 kDa, which exhibited immunostimulatory activity. Thus, the temperature is an important factor influencing the yield and properties of SGs. It can be seen that cold water extraction avoids the possible damage of high temperatures during extraction, which might contribute to maintaining the activity of SGs.

### 2.3. Assistance Extraction

A combination of conventional extraction methods with other treatments, such as ultrasound, microwave, and enzymes, could save time during extraction process, as well as reduce the solvent volume and the energy needed [78].

#### 2.3.1. Ultrasonic-Assisted Extraction (UAE)

Ultrasonic waves damage cell walls and increase the dissolution of organic compounds in cells [47]. Shi et al. [65] explored the UAE method with 50 W of ultrasonic power at 87 °C for 31.7 min by using an ultrasonic-microwave apparatus (CW-2000, Shanghai Xintuo Microwave In-strument Co., Ltd., Shanghai, China) to obtain *G. lemaneiformis* SGs with a high yield of 34.8%. Moreover, high-power ultrasound resulted in polysaccharide degradation. Yu et al. [64] conducted ultrasonic treatment for *P. yezoensis* SGs was carried out by using a HF-20B ultrasonic reactor (Beijing Hong Xiang Long Biotech-nology Developing Co., LTD, Beijing, China) under 400–1200 W and pulsed as 2–8 s/2 s (on/off) at 30 °C for 4 h to degrade large molecules and obtain products in a narrower molecular weight distribution. Moreover, prolonged exposure to ultrasound may change the microstructure of polysaccharides and affect their bioactivity [72]. Thus, the processing control of UAE should be inspected for the quality assurance of SGs.

#### 2.3.2. Microwave-Assisted Extraction (MAE)

MAE is another extraction method which significantly improves yield and saves time and energy [79]. Chen and Xue [80] compared the efficiency of MAE from microwave assisted extraction equipment (NJC 03-2, 2450MHz, Nanjing, China) with the traditional hot water extraction (THWE) for SGs from *Porphyra haitanensis*. MAE was carried out with a ratio of water to raw material of 28.98 mL/g as well as 77.84 W of microwave power for 14.14 min. Compared with THWE, MAE greatly enhanced the yield from 1.72% to 5.01% and reduced extraction time from 300 min to 14 min. Moreover, the structure analysis of SGs by FTIR spectra showed no changes in characteristic peaks. The study indicated that MAE was a productive technique for SG extraction.

#### 2.3.3. Enzyme-Assisted Extraction (EAE)

More recently, EAE technology has been widely used to improve the yield of polysaccharides owing to its lower cost, low energy consumption, and eco-friendliness when compared with traditional methods [81]. Thus, extraction conditions, such as extraction time, extraction rate, seaweed/solvent ratio, temperature, pH, and extraction solvent, also have influenced the efficiency of enzyme extraction and have been investigated to obtain the optimal yield [25,30,31,50,82]. Cavalcante Alencar et al. [83] improved SG extraction from *G. caudata* by papain digestion, which resulted in a high yield of 24.96% of SGs with no changes in the structure, furtherly proving by FTIR spectra (Shimadzu 204 IR spectrophotometer (model 8300), Kyoto, Japan). Coura et al. [84] also used a similar extraction method of papain (Sigma Aldrich, St. Louis, MO, USA) digestion to extract SGs from *G. cornea*, resulting in a yield of 18% and a sulfate content of 15.66%. A similar method was applied in SG extraction from *G. biraiae* by Silva et al. [85]. Apart from papain which was a protease, cellulase was also adopted in SG extraction. Zhang et al. [86] treated *A. plicata* with cellulase of 10 U/kg at 50 °C for 4 h to isolate crude SGs.

Xiao et al. [81] compared three SGs of traditional alkali-extracted agar (AA), enzyme-extracted agar (EA), and enzyme-assisted extracted agar (EAA), which were prepared from *G. lemaneiformis* by using alkaline solution, cellulases and arylsulfatase, respectively. The result indicated that the EA sample showed the highest yield, while the AA sample showed the lowest. Likewise, the AA sample had a lower sulfate content (0.84%) than those of EAA (1.39%) and EA (4.45%). Fidelis et al. [23] combined the efficiency of sonication and enzymatic digestion, which resulted in a significant increase in the yield and variation in antioxidant and anticoagulant activities. Hence, the EAE method, whether or not in combination with other assistant methods, has been used for SG extraction in various studies, which can be applied to obtain SGs on an industrial scale.

Different pretreatment methodologies produced SGs with different compositions and characteristics. Al the available data were summerized and compared in Table 1. 

## 3. Purification and Structure Features of SGs

### 3.1. Purification

Currently, column chromatography is the most widely used method to purify polysaccharides due to its excellent purification efficiency [82,89]. As shown in Table 1, SGs were fractionated by the anion exchange chromatography (DEAE-52 cellulose, DEAE-Sepharose fast-flow, Q-Sepharose Fast Flow, etc.) and the gel-filtration chromatography (Sephadex G-100, High-Resolution Sepharose 4-LB, etc.) according to the composition of acidic or neutral groups and molecular sizes. For example, crude SGs from *G. rubra* were purified by the DEAE-52 cellulose column and Sephadex G-50 column to obtain three fractions of GRPS-1-1, GRPS-2-1, and GRPS-3-2 with sulfated contents of 5.96, 8.46 and 12.03% and average molecular weights of 1310, 691, and 923 kDa, respectively [30]. Zhang et al. [86] purified SGs from *A. plicata* with three methods, polyethylene glycol (PEG) precipitation, DEAE-cellulose chromatography, as well as the combination of the two methods above, to obtain three fractions, i.e., PA, DA, and PDA, with yields of 13.86, 14.48, and 10.44%, respectively. PDA fraction showed the lowest sulfated content of 0.07%, while those of PA and DA were 0.28 and 0.15%, respectively. The biological activities of SGs depend on the structure features of molecular size, type of sugar, and sulfated contents, as well as the type of linkage, molecular geometry, and sulfated position; thus, the purification is a fundamental process to exploring the fine structure and elucidating the relationship between the fine structure and biological activity.

### 3.2. Structure Features

The structure features of SGs from agarophytes of *Gracilaria* spp. [29,45,70], *Gelidium* spp. [26], *Porphyra* spp. [53,75,93] and *Ahnfeltia* spp. [16,91] are compared as follows.

#### 3.2.1. *Gracilaria* spp.

A considerable amount of SGs from *Gracilaria* spp. exist in the form of galactans. In general, SGs from *Gracilaria* spp. were composed of 3-linked β-d-galactopyranose (G) and 4-linked 3,6-anhydro-α-l-galctopyranose (LA) with sulfate groups in C-4 of D-galactopyranose, C-6 of D-galactopyranose, and C-6 of l-galactopyranose. Their typical structures are shown in Figure 3 [45,94,95]. Galactan polysaccharides of agars refer to lconfiguration, while carrageenans refer to d-configuration. Additionally, 3,6-anhydro derivatives may result in partially or totally 4-linked residues. Thus, repeating units as hydroxyl groups may be substituted by ester sulfate, methyl group, pyruvic acid, and single monosaccharides substitution (d-xylose, d-glucuronic acid, 4-o-methyl-l-galactose,) with a varied position of different species [94,96].

#### 3.2.2. *Porphyra* spp.

SGs in *Porphyra* species are named Porphyran. As shown in Figure 4, it is comprised of a linear backbone of alternating 3-linked β-d-galactosyl (G) units and 4-linked α-l-galactosyl 6-sulfate (L6S) or 3,6-anhydro-α-l-galactosyl (A) units, while it is distinctive in terms of having L6S [64,90,97,98]. In addition, the galactan backbone is modified by methyl ether groups (M) at the C2 position of A units to form A2M residues or at the C6 position of G units to form G6M residues [99]. SG from each *Porphyra* spp. species has its own chemical features which might be influenced by the growing conditions [100]:

#### 3.2.3. *Gelidium* spp.

SGs from *Gelidium* species are characterized as agar-type galactans containing residues of 3,6 anhydro-α-l-galactopyranose and a variable sulfation pattern (β-d-galactopyranose-2-sulfate-O-α-d-galactopyranose-2,3 sulfate [101,102] (Figure 5). SGs from the *Gelidium* spp., which are composed of 2,3-disulfated galactose units, exhibited anticoagulant activity, which might be due to the presence of the modificated sulfate groupat galactose units. The different derivatives at the fourth and sixth carbon positions are the carboxyl group and sulfated group, respectively [15].

#### 3.2.4. *Ahnfeltia* spp.

SGs from *Ahnfeltia* are also be ascribed to agaran-type polysaccharides by normal and second-derivative spectra from FTIR results at 931.8, 893.6, 790.9, 771.3, 741.0, 717.7, and 693.9 cm^−1^ [16], which composed of alternating 3-linked β-d-galactopyranosyl and 4-linked α-l-galactopyranosyl or 3,6-anhydro-α-l-galactopyranosyl residues. Methyl disaccharide units are present in position two of 3,6- anhydro-l-galactose (LA) residues and absent in d-galactose residues (G) [92] (Figure 6).

### 3.3. Structure Difference among Four Genera

The sulfated polysaccharides from four different species of agarophytes, *Gracilaria* spp., *Gelidium* spp., *Porphyra* spp., and *Ahnfeltia* spp., expose a similar linear backbone but different derivative positions with variable sulfation patterns. Most of their linear backbones consist of two main units of 3-linked β-d-galactose units and 4-linked α-l-galactose units. The carbon 6 of the galactose units is very active and can be modified by sulfated groups as well as other substituents, i.e., methyl ethers, acetates, and pyruvic acid ketals [14]. Significantly, SGs from different species vary in the modification pattern and the proportion of 3,6-anhydrogalactose. Thus, it is crucial to identify the repeating units and substituts [94,103,104]. The sulfated group is found to be more in *Gelidium* spp. and *Gracilaria* spp. agars, and the methylate groups and pyruvic acid ketals are commonly found in *Gracilaria* spp. agars. 

Furthermore, some species of *Gelidium* spp. and *Porphyra* spp. were revealed to contain agar composed mainly of 6-OCH_3_-agarobiose with high contents of methoxyl groups [105]. *Ahnfeltia* spp. has the same principal backbone as other agarophytes, but the difference belongs in the methyl group in the C-2 position of 1,4-linked-α-l-anhydro-galactose. Some research revealed that SGs of *Porphyra* spp. comprises alternating 3-linked β-d-galactose, 4-linked α-l-galactose6-sulfate, and 3,6-anhydro-a-l-galactose units on their linear backbone, in which their C-6 position of the d-galactose units are found to be high-degree methylated [106].

In comparison, SGs from *Gracilaria* spp. have higher degrees of sulfation, methoxylation, and pyruvylation than those from *Gelidium* spp. [107]. The complex mixture of SG units had a varying structure because various hydroxyl groups within the biopolymer could be substituted by sulfur ester, methyl ester, pyruvate acetal, or single-branching sugar residues acid, with their position varying from species. Thus, the physical, rheological, and bioactivity properties may possibly be affected by structure changes [15,26,108]. Moreover, the chemical properties of SGs from different species also show significant variations owing to their molecular weights. The information on reported SGs is shown in Table 1. The location and proportion of these substituents in SGs strongly affect the physiochemical properties. One impressive characteristic of soluble behavior was strongly affected by sulfation patterns [14]. As shown in the table, most of the biological properties of SGs were associated with the presence of sulfated groups.

## 4. Bioactivity of Sulfated Galactans

Based on more and more in-depth research, SGs are recognized as healthy attributes aside from food sources, indicating that they have the potential to be applied as natural functional ingredients, such as in functional foods, pharmaceuticals, and cosmetics, for industries [13,60]. Data on the biological activities of SGs from agarophytes are summarized in this review.

### 4.1. Anti-Cancer and Anti-Tumor Activity

Cancer is an abnormal growth of tissue or cells exhibiting uncontrolled division autonomously, resulting in a progressive increase in the number of cell divisions [109]. There are many chemotherapeutic agents used for cancer curing, which depend on the type and stage of cancer. Treatments to eradicate the tumor or slow its growth may include some combination of surgery, radiation therapy, chemotherapy, hormone therapy, or immunotherapy [110]. However, due to the considerable side effects and resistance of cancer cells to these drugs, it is important to discover natural products as effective anti-cancer agents with low toxicity to battle the disease [19,21]. Thus, there are various studies on natural alternative materials to be used instead of the present remedy, and seaweed can be one such alternative.

As listed in Table 2, various in vitro and in vivo studies have been carried out to discover SGs for anti-cancer or anti-tumor activity. SGs from *G. edulis* exhibited greater anticancer activity against human lung adenocarcinoma cell line A549 when compared with the standard drug Cisplatin, which showed no adverse effects, confirmed in both in vitro and in vivo models [19]. It was safe to use *G. edulis* orally, up to the maximum treatment dose of 1000 mg/kg as an anti-lung cancer agent. Erfani et al. [110] concluded that ethanol extract of *Gracilaria foliifera* showed the best anticancer activity against human breast cancer cell lines among ten algae of the Persian Gulf and Oman Sea with the estrogen receptor/progesterone receptor-independent mechanism for their cellular growth inhibition, which was an excellent candidate for further purification to obtain novel anticancer substances. SGs of *G. fisheri* from Thailand was also used to investigate its anti-proliferation effect on cancer cell by Sae-lao et al. [20,21]. The results showed that the SGs could inhibit the proliferation of Cholangiocarcinoma Cells (CCA), and its mechanism of inhibition was mediated by inhibited EGFR activation and the EGFR/ERK signaling pathway. Moreover, SGs could reduce the migration rate of CCA and block the phosphorylation of EGFR and extracellular signal-regulated kinase. Thus, SGs from *G. fisheri* were suggested as potential agents for therapeutic prospects in CCA treatment.

Extraction and purification methods affected the antitumor activity of SGs. SG fraction of GLP-3, which was extracted from *G. lemaneiformis* by ultrasonic microwave and purified by DEAE-Sephadex A-50 column, showed the highest anti-tumor activities than those of other fractions [65]. Yu et al. [64] compared the antiproliferative effects of ultrasonic pretreated (U-PP) and non-treated (PP) SGs from *Porphyra yezoensis* on the degradation and inhibition of cancer cell lines. U-PP showed higher activity than PP in the same dose, which indicated the possible modification of SGs by ultrasonic treatment. The conjugation with low-Mw Porphyran could reduce the undesirable side effects of the anticancer drug 5-Fluorouracila [111]. They found that low-Mw Porphyran carrying 5-fluoracil could enhance antitumor activity by increasing the inhibition rate and TNF-α and NO secretion and improve the immune system of the transplanted S180 tumor mice.

### 4.2. Anti-Coagulation Activity

Anti-coagulation activity is the most widely studied activity of sulfated polysaccharides. All data on the anticoagulant activity of SGs are collected in Table 3. Pereira et al. [15] isolated and purified an SG from *G. crinale* and explored its anticoagulant activity. Activated partial thromboplastin time (APTT) assay indicated that the SG showed almost coextensive curved thrombin inhibition in the presence of anti-thrombin. Additionally, SG from *G. amansii* was investigated for anticoagulant activity using both in vitro and in vivo methods [112]. The results indicated that it could effectively prolong the coagulation time (CT), prothrombin time (PT), thrombin time (TT), and APTT in a dose-dependent manner in vitro by restraining both extrinsic and intrinsic coagulation pathways. The in vivo results showed that the oral administration of SG could inhibit the coagulation process in rats by prolonging TT and PT. Thus, SG from *G. amansii* could be used to develop new drugs for blood-coagulated healing. Crude and purified SGs from *G. debilis* presented discreet inhibitions of coagulation by both ATPP and PTT assays, and the activity was co-related with high sulfated content and low molecular weight [24].

Fidelis et al. [23] found that a combination of extraction procedures of SGs from *Gracilaria birdiae* affected its anticoagulant activity, among which the high molecular weight (>30 kDa) SG extracting by water at 22 °C, sonication at 60 °C, and proteolysis at 60 °C showed the best activity. SGs from *Porphyra perforata* were extracted by both organic and aqueous solvents, and the aqueous extracts showed better anti-coagulation activity through the intrinsic pathway [113]. Zhang et al. [87] revealed that the anticoagulant activity referring to the prolonged APTT, TT, and PT of SGs from *P. haitanensis* strongly depended on the position of the sulfated group. Matsuhiro et al. [16] revealed that SGs isolating from *A. plicata* and treated with the sulfation process (SO_3_-pyridine complex) exhibited an APTT value of 1.36 ± 0.04, which was notably similar to that of heparin. Additionally, the antioxidant capacity of the crude SG extracted in a low alkaline solution (NaBH_4_) revealed great activity of the oxygen radical absorbance capacity (ORAC) value similar to that of a commercial κ-carrageenan. What is more, the sulfation process exhibited SGs with better properties. The result indicated that SGs from *A. plicata* possessed both anti-coagulation and anti-oxidation activities.

### 4.3. Anti-Inflammatory Activity

As shown in Table 4 and Figure 7, various in vitro and in vivo studies have validated the anti-inflammatory potential of SGs from different agarophytes. SGs from *Gracilaria cornea* showed anti-inflammatory and antinociceptive activities by reducing nociceptive responses, inhibiting paw edema induced by carrageenan and dextran in mice, and increasing spleen weight without any injuries in mice organs [84]. SGs from *Gracilaria caudata* could also reduce carrageenan-induced paw edema and decrease levels of inflammation cytokines mediators of TNF-α and IL-1β [114]. Coura et al. [115] revealed that SGs from *G. cornea* exhibited anti-inflammatory activities by reducing mRNA and protein levels of IL-1β, TNF-α, and COX-2 in cells. SGs from *G. cornea* could reduce NO_2_/NO_3_ levels, increase GSH levels, decrease lipid peroxidation, as well as modulate the transcription of neuroprotective and inflammatory genes in the rat model of 6-hydroxydopamine-induced Parkinson’s disease [116]. Oliveira et al. [27] reported the SGs from *G. caudata* effectively reduced the inflammatory disorders in an arthritis model. Dutra et al. [28] also found that the treatment with SGs from *G. caudata* reduced the wet weight of the colon, inflammatory infiltrate, oxidative stress, pro-inflammatory cytokine action, and iNOS expression in mice.

Isaka et al. [25] found that discolored waste nori from *P. yezoensis* showed better anti-inflammatory activity on LPS-induced RAW 264.7 cells than that of normal nori. After that, the in vivo activity was proved by Nishiguchi et al. [117] in mice by decreasing the malondialdehyde level in mice livers. Yanagido et al. [94] further investigated the anti-inflammatory activity of degraded porphyrans of D1-D4 with decreased molecular size, among which D2-porphyran exhibited the highest inhibitory activity against NO production and TNF-α secretion. Likewise, the similar activity of porphyran from *P. yezoensis* was revealed by Jiang et al. [68]. Furthermore, Ueno et al. [118] revealed that the anti-inflammatory mechanism of SGs from *P. yezoensis* was caused by the downregulation of NF-κB in RAW 264.7 cells.

Purified SGs from *G. crinale* showed in vivo anti-inflammatory activities by the inhibition of paw edema induced by both histamine and dextran [119]. SGs from *G. amansii* showed anti-inflammatory activities in LPS-induced RAW 264.7, and the mechanisms were reported by Kim et al. [120]. Cui et al. [26] obtained purified SGs from *G. pacificum* Okumura and investigated the anti-inflammatory in LPS-stimulated human monocytic (THP-1) cells. Based on in vitro and in vivo studies, as well as the clearly elucidated mechanisms, SGs from different agarophytes show great potential to be developed as anti-inflammatory agents. More human studies are urgently required for their further application.

**Table 4 biomolecules-13-01745-t004:** Summary of anti-inflammatory effects of SGs from seaweed of agarophytes.

Seaweed Species	Main Compounds	Model Method (Cell Lines)	Proposed Mechanism of Action	Reference
*Gelidium amansii*	Sulfated polysaccharides	LPS-induced RAW 264.7 macrophage cell	Increased protein expression, suppressed the transcription factors for adipogenesis, suppressed the production of TNF-α ^1^, increased JNK ^2^ expression, inhibited differentiation of pre-adipocytes, and increased the quantity of glycerol released	[120]
*Gelidium crinale*	Sulfated galactans	Several inflammatory agents induced paw edema and intravenous (i.v.) route in rodent experimental model	Inhibited the time course of dextran-induced paw edema, inhibited the paw edema induced by histamine, inhibited both neurogenic and inflammatory phase of the formalin test, and well tolerated by animal	[119]
*Gelidium pacificum* Okamura	Sulfated polysaccharides	LPS-stimulated human monocytic (THP-1) cells	Reduced NO ^3^ production and suppressed the mRNA and protein expression of TLR-4 ^4^, MyD88 ^5^, and TRAF-6 ^6^	[26]
*Gracilaria caudata*	Sulfated polysaccharides	Several inflammatory agents (carrageenan, dextran, bradykinin, and histamine) induced paw edema and peritonitis in Swiss mice	Reduced the carrageenan-induced paw edema, the levels of TNF-α and IL-1β ^7^, cytokine levels in the peritoneal cavity, and hypernociception	[114]
*Gracilaria caudata*	Sulfated polysaccharides	Acetic acid-induced intestinal damage in UC model in mice	Reduced the macroscopic and microscopic scores, wet weight of colon, MPO ^8^ enzyme activity, GSH ^9^ consumption, MDA ^10^, NO_3_/ NO_2_ and pro-inflammatory cytokine (IL-1β and TNF-α) concentrations, iNOS ^11^ expression, and oxidative stress	[28]
*Gracilaria cornea*	Sulfated polysaccharides	Carrageenan-induced rat paw edema and dextran-induced rat paw edema	Reduced nociceptive response, reduced licking time in both phases of the formalin testand no systematic damage on rat’s body	[84]
*Gracilaria cornea*	Sulfated polysaccharides	Several inflammatory agents (carrageenan, dextran, serotonin, bradykinin, compound 48/80, or L-arginine) induced paw edema in mice	Inhibited rat paw edema induced by histamine and inhibited IL-1β, TNF-α, and COX-2 ^12^ mRNA and protein level	[115]
*Gracilaria cornea*	Sulfated polysaccharides	Rat 6-hydroxydopamine Parkinson’s disease model	Reduced NO_2_/NO_3_ levels, increased GSH level, decreased lipid peroxidation, and modulated the transcription of neuroprotective and inflammatory genes	[116]
*Porphyra yezoensis*	Sulfated polysaccharides (Porphyran)	LPS-induced RAW 264.7 macrophage cell	Inhibited NO production, the expression of inducible nitric oxide synthase (iNOS), and the LPS-induced NF-iB ^13^ activation	[68]
*Porphyra yezoensis*	Sulfated polysaccharides	LPS-induced endotoxin shock in mice	Reduced the production of NO and TNF-α and decreased the malondialdehyde level in the liver	[117]
*Porphyra yezoensis*	Sulfated polysaccharides	LPS-induced RAW 264.7 macrophage cell	Inhibited the secretion of NO and TNF-α (low molecular size fraction)	[94]
*Porphyra yezoensis*	Sulfated polysaccharides	RANKL-induced differentiation of RAW 264.7 cells into osteoclasts	Suppressed the osteoclastic differentiation, the formation of RANKL ^14^ -induced multinucleated giant cell, and the mRNA expression of osteoclastogenesis-related genes with no cytotoxicity	[118]

^1^ Tumor necrosis factor- α. ^2^ c-Jun N-terminal kinase. ^3^ Nitric oxides. ^4^ Toll-like receptor. ^5^ Myeloid differentiation primary response gene. ^6^ Tumor necrosis factor receptor-associated factor 6. ^7^ Interleukin 1 beta. ^8^ Myeloperoxidase. ^9^ Glutathione. ^10^ Malondialdehyde. ^11^ Inducible nitric oxide synthase. ^12^ Cytooxygenase inducible isoform. ^13^ Nuclear factor-kappaB. ^14^ Receptor activator of nuclear factor kappa-B ligand.

### 4.4. Anti-Lipidermic, Anti-Obesity, and Anti-Diabetic

Obesity, one of the greatest health concerns, was recently proven to be associated with various gastrointestinal diseases and symptoms [121]. Various in vitro and in vivo models indicated the efficiency of SGs from *Gracilaria* spp. and *Phorphyra* spp. as anti-lipidermic, anti-diabetic, and anti-obesity agents without any side effects (Table 5). The mechanisms of anti-lipid activities are shown in Figure 7. SGs from *G. lemaneiformis* and *G. opuntia* were explored for diet-induced obesity in many studies. Sun et al. [38] indicated that dietary supplements with SGs from *G. lemaneiformis* reduced weight gain, total cholesterol (TC), and low-density lipoprotein cholesterol (LDL-C) contents, as well as decreased fat accumulation in the liver and adipose tissues. In addition, SGs improved the body composition of high-fat diet-fed (HFD) mice by increasing the colon length and repairing the damaged intestine, which was due to the promoted utilization of polysaccharides and lipids. Li et al. [34] further discovered that SGs from *G. lemaneiformis* regulated lipid metabolism and accelerated free fatty acid oxidation by upregulating the expression of the PPARα, ACS, and CPT1a genes. Makkar et al. [41] reported that SGs from *G. opuntia* showed antidiabetic activities through the inhibition of various related enzymes and suppressed the hyperglycemic response.

SGs from *P. yezoensis* could improve glucose metabolism in diabetes by upregulation of adioinectin levels, increasing short-chain fatty acids contents, and improving gut health in HDF-induced diabetic mice [17,35]. SGs from *P. haitanensis* could improve serum lipid levels and alleviate damaged tissue in diabetic mice, which could be developed as a potential oral hypoglycemic drug [36,37]. Tsuge et al. [32] discovered that SGs from *P. yezoensis* also affected lipid metabolism by decreasing the food intake, body weight gain, renal adipose tissue weight, and serum cholesterol levels in rats, and SGs of higher sulfate content (HS-POR) were more effective than those of the lower sulfate content (LS-POR).

### 4.5. Anti-Virus and Anti-Microbial Activities

In aquaculture, microbial contamination is one of the causes of diseases in aquatic animals like fish and shrimp, especially the genus of *Vibrio*, which results in high mortality in aquaculture. White spot syndrome virus (WSSV) is the most damaging pathogen in terms of production and economic losses for worldwide-shrimp farming. Rudtanatip et al. [42] revealed the antiviral activity of SGs from *G. fisheri* against WSSV infection in *Penaeus monodon* (black tiger shrimp) by decreasing the cytopathic effect (CPE) and cell mortality through binding the protein of the virus. They further investigated that an SG-supplemented diet stimulated antibacterial activity and enhanced shrimp immunity for the prevention of bacterial diseases [43]. Similar results were reported by Wongprasert et al. [45]. Anti-WSSV infection in juvenile *Litopenaeus vannamei* (Pacific white shrimp) was studied by Afsharnasab et al. [44] and Cantelli et al. [122], who reported that SGs could enhance the survival rate and immune parameters. Amorim et al. [88] discovered that SGs from *Gracilaria ornata* could inhibit only the growth of *E. coli* but not other bacteria such as *Bacillus subtilis*, *Staphylococcus aureus*, *Enterobacter aerogens*, *Pseudomonas aeruginosa*, *Salmonella choleraesuis*, and *Salmonelatyphi*.

Diarrheagenic *E. coli* can enter the gastrointestinal digestive system through food and adhere to and damage the gastrointestinal tracts of humans (Table 6). Liu et al. [53,54] reported the antibacterial activity against *E. coli* of SGs from *G. verrucosa*. They exhibited the activity through the disruption of the cell membrane structure; meanwhile, the low-MW SGs penetrated the cell walls and eventually reached the interior of *E. coli*. Thus, based on the revealed anti-virus and anti-microbial activities, SGs from agarophytes could be applied as natural drugs for aquatic animals and potential therapeutic agents for humans.

### 4.6. Immunostimulating Activities

Immune mediators released from macrophages play significant roles in regulation of immune responses [102]. Several studies indicated the immunostimulating activity of SGs in RAW 264.7 macrophage and mice models (Table 7). Immuno-efficiency of SGs from *Porphyra* spp. was investigated. Bhatia et al. [18] indicated that SGs of *P. vietnamensis* enhanced various haematological parameters and levels of white blood cell of Wistar albino rats and albino mice by oral administration. Liu et al. [66] investigated in vitro (RAW 264.7 cell) and in vivo (BALB/c murine model) immunomodulatory activities of SGs from *P. haitanensis* (PHPS), which indicated that the purified PHPS increased the phagocytosis activity and enhanced the secretion of IL-6, IL-10 and TNF-α in cell. Moreover, PHPS promoted the proliferation of mice lymphocytes and induced the generation of TNF-α and IL-10. Ren et al. [47] found that the low molecular weight SG (GLP) of *G. lemaneiformis* could improve the proliferation and pinocytic capability of RAW 264.7, and promote the production of ROS, NO, IL-6, TNF-α without cytotoxicity.

Moreover, Di et al. [30] indicated a higher immunomodulatory activity of the purified GRPS-3-2 fraction than those of crude polysaccharides and other fractions, which might due to its higher sulfated content. Food allergy incorporates a variety of immune-mediated adverse reactions to foods that occur in genetically predisposed individuals. Liu et al. [123] reported the anti-allergic properties of purified SG (GLSP) from *G. lemaneiformis* in both cell line model (RBL-2H3 and KU812) and animal model (BALB/c mice). SGs from *P. haitanensis* suppressed Th2 immune responses by modulating their imbalance and thus possessed anti-allergic activity in mice by ways of both intraperitoneal injection and oral therapeutic administration [46]. Another research from Han et al. [124] showed that SGs extracted from both *G. lemaneiformis* and *P. haitanensis* exhibited anti-allergic activities in mice through attenuating OVA-induced anaphylaxis by upregulation of the contents of regulatory T cells. Thus, in vitro and in vivo studies revealed the immunostimulating activities and mechanisms of SGs from agarophytes.

### 4.7. Gut Microbiota Regulation

Prebiotics are non-digestible carbohydrates that act as “food” for probiotics. Attention has been paid to the application of polysaccharides from marine algae as prebiotics, which serve as dietary fibers that are indecomposable by enzymes of the upper gastrointestinal tract [125]. The studies of the prebiotic activities of SGs are listed in Table 8. 

The results showed SGs could not be digested under upper gastrointestinal tract conditions but could slowly degrade and be utilized by gut microbiota, which leads to the increase in SCFA concentrations, especially acetic, propionic, and isobutyric acid, modulating the microbial community. SGs from *G. pacificum* Okamura had beneficial effects on mice with antibiotic-associated diarrhea (AAD) by increasing the richness and diversity of the positive gut microbiota such as *Bacteroides*, *Oscillospira*, and *Bifidobacterium*, improving the content of SCFAs, downregulating the concentrations of inflammatory cytokines, and improving metabolic pathway [48]. Han et al. [49] also found that SGs from *G. lemaneiformis* could hardly degrade when passing through the simulated digestion system, while *Desulfovibrio* and *Bacteroides* were dominant genera regarding the utilization of SGs. Di et al. [126] also investigated the gut microbiota modulation effects of SGs from *G. rubra* in mice. Thus, current studies indicated SGs as new potential prebiotics which can be used in functional foods.

### 4.8. Anti-Diarrhea

Diarrhea has many different causes, such as the consumption of contaminated food and water, inflammatory diseases, and infection by pathogens such as *Vibrio cholerae* [127]. Presently, there is no effective pharmacological cure; thus, patients only receive oral rehydration therapy to reduce the likelihood of disease development [50]. Several studies focused on the anti-diarrhea activity of SGs to develop new therapeutic agents from natural products (Table 9). Silva et al. [85] suggested that SGs from *G. birdiae* exhibited anti-diarrhea activity by the inhibition of inflammatory cell infiltration and lipid peroxidation against naproxen-induced gastrointestinal damage in rats, which supported their development for the prevention of non-steroidal anti-inflammatory drugs (NSAID)-encouraged adverse effects in the gastrointestinal tract in humans. Costa et al. [50] indicated that SGs from *G. caudata* possessed anti-diarrhea activity by interacting with the Cholera toxin and monosialoganglioside GM1 receptor in the induced diarrhea model of Wistar rats. Bezerra et al. [51] also reported the significant antidiarrheal effect of SGs from *Gracilaria cervicornis* using gastrointestinal disordered Swiss mice. Two SGs, PHSP (hp) and GLSP (hp), were extracted by Liu et al. [52] from *P. haitanensis* and *G. lemaneiformis*, with an average molecular weight of 165.45 kDa and 71.87 kDa (mainly), respectively. Both SGs showed anti-diarrhea activity by specific and non-specific immune regulations in mice. Compared with that of the LPS group, the OD_540_ indicated the activity of nitroblue tetrazolium (NBT) of diarrhea in serum was significantly decreased upon treatments by PHSP (hp) and GLSP (hp). What is more, GLSP (hp) showed better activity than PHSP (hp) because of its lower OD_540_ value, which indicated that SGs with a lower molecular weight might possess better anti-diarrhea activity. These findings provided evidence for the use of SGs from agarophytes as functional food agents to mediate and alleviate diarrhea.

## 5. Conclusions and Future Directions

Marine macroalgae, in the special seawater environment of the habitat, have been explored as natural biomass by the functional food and pharmaceutical industries to produce new food and novel drugs for the improvement of human health. In the last two decades, studies on the isolation, structural identification, and bioactivities of SGs from agarophytes have steadily increased. SGs have demonstrated broad biological activities, i.e., anti-tumor, anticoagulant, anti-inflammatory, antioxidant, anti-obesity, anti-diabetic, anti-microbial, anti-diarrhea, and gut microbiota regulation properties, which have indicated that SGs have promising potentials as health improving agents. The collected evidence has indicated that variations in biological activities depend on various structural features of SGs, including molecular mass, the composition of monosaccharides, the degree of sulfated esterification, the position of sulfated esterification, and the degree of polymerization, which have been affected by geographical conditions (salinity, temperature, light) and preparation methods (pretreatment, isolation, and purification methods).

Despite the increasing efforts conducted by scientists, to date, there is no application of SGs in the nutraceutical and pharmaceutical industries. This review demonstrates three approaches for developing SGs into products with high economic value. Firstly, the influence of taxonomy and ecological factors, including growth environmental factors and harvest period, on SGs should be comprehensively studied. Steady aquaculture and a clear understanding of biomass are fundamental for obtaining various types of SGs. Secondly, the relationship between a fine structure and the bioactivity of purified SGs from agarophytes is needed. Several published papers did not further identify the SGs in which crude extracts were used for activity studies. Thirdly, clinical trials of SGs should be carried out in the future. Currently, bioactivity studies are still in the preliminary stage using cells and animal models, while clinical settings are not enough to support their real impacts on human beings. The agarophytes have well-established aquaculture fundamentals, which support their stable biomass production. Thus, further study of SGs according to the suggestions above can benefit the development of the seaweed resource and promote the Blue Economy.

## Figures and Tables

**Figure 1 biomolecules-13-01745-f001:**
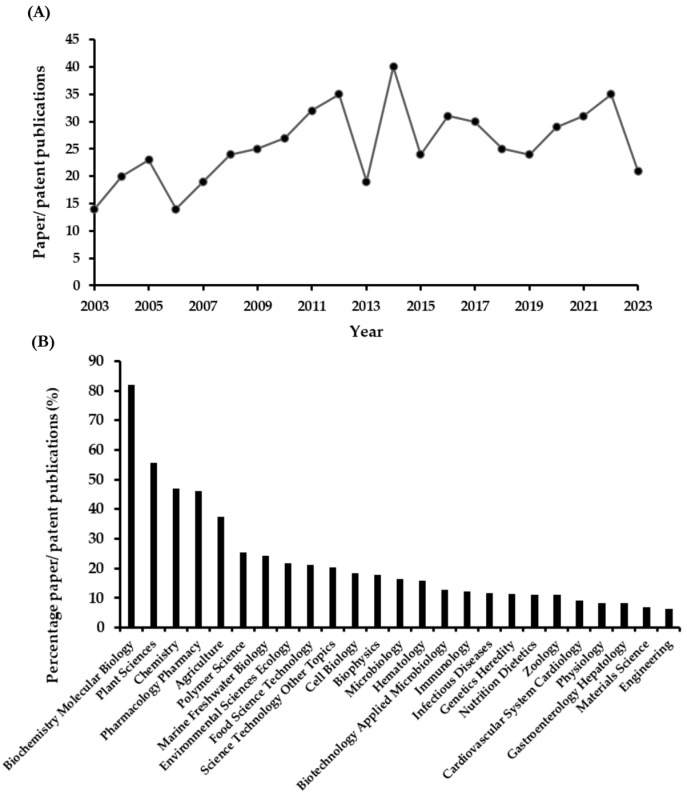
Statistical analysis of published papers and patents related to SGs within the past 20 years. (**A**) Annual publication. (**B**) Research areas.

**Figure 2 biomolecules-13-01745-f002:**
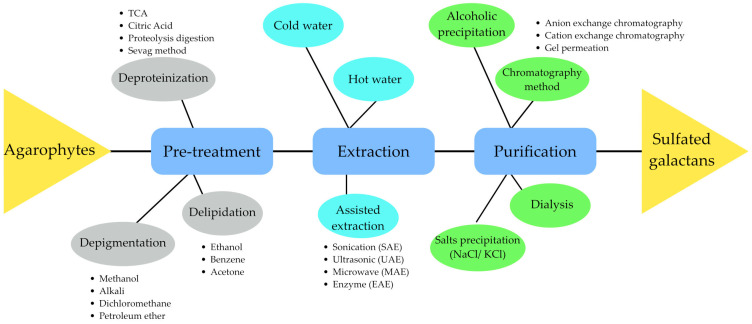
The procedure and methods of pretreatment, extraction, and purification of sulfated galactans.

**Figure 3 biomolecules-13-01745-f003:**
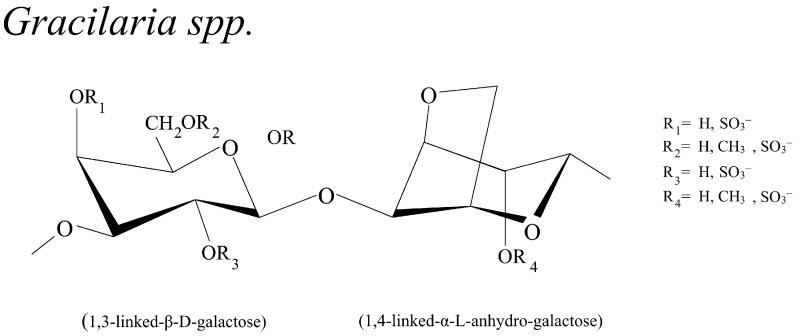
Schematic representation of sulfated galactans from red seaweed, *Gracilaria* spp. [45,95].

**Figure 4 biomolecules-13-01745-f004:**
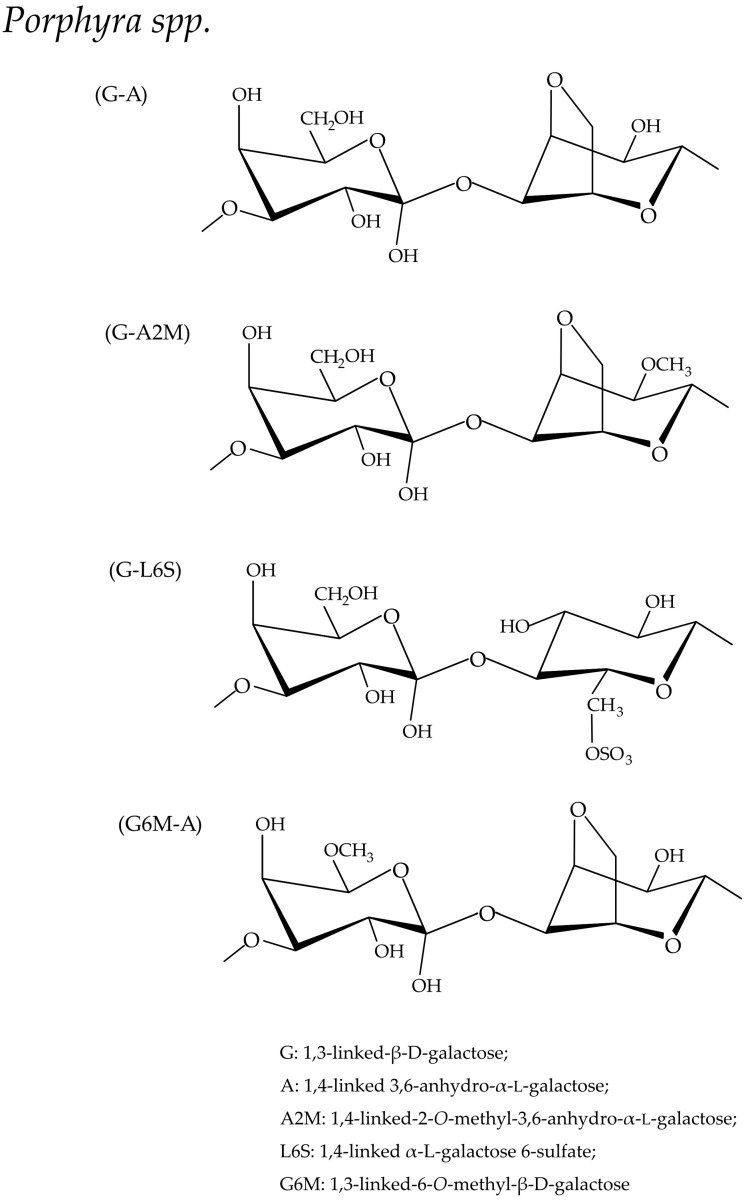
Schematic representation of sulfated galactans from red seaweed, *Porphyra* spp. [64].

**Figure 5 biomolecules-13-01745-f005:**
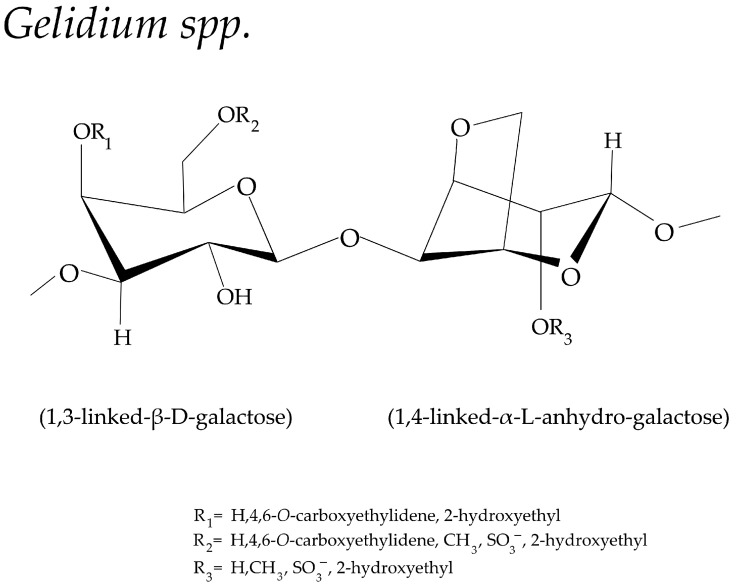
Schematic representation of sulfated galactans from red seaweed, *Gelidium* spp. [15,103].

**Figure 6 biomolecules-13-01745-f006:**
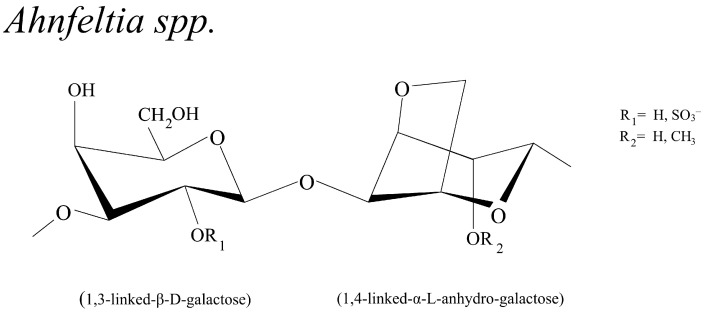
Schematic representation of sulfated galactans from red seaweed, *Ahnfeltia* spp. [16].

**Figure 7 biomolecules-13-01745-f007:**
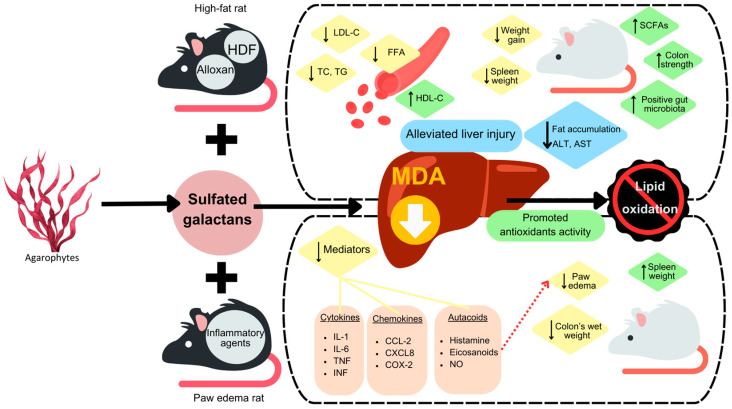
Overview of related bioactivities of anti-inflammatory and anti-lipid of SGs from seaweed of agarophytes. Sulfated galactans (SGs) express the main potential to decrease malondialdehyde (MDA) levels in liver, which affects the capacity of lipid metabolism and the release of pro-inflammatory mediators such as cytokines and chemokines, including autacoids. When MDA level is decreased, the liver is alleviated from injury from improper lipid and inhibits lipid oxidation. SG treatment shows improvements by these mechanisms in the induced rats model.

**Table 1 biomolecules-13-01745-t001:** A summary of chemical and physical compositions and bioactivity of SGs obtained from agarophytes (*Gelidium* spp., *Gracilaria* spp., *Porphyra* spp., *Ahnfeltia* spp.).

Agarophytes	Species	Extraction, Fraction, and Purification	Molecular Weight (kDa)	Yield (%)	Protein Content (%)	Total Sugar (%)	Sulfated Content (%)	Monosaccharide Composition	Biological Activity	Reference
*Gelidium* spp.	*G. pacificum* Okamura	Ultrasound extraction (80 °C), fractionation (DEAE-52 cellulose column), and purification (Sephadex G-100 gel column); GPOP-1	28.807	-	1.14	69.27	8.8	86.2% (Gal) ^1^, 10.26% (Xyl) ^2^, 1.35% (Glu) ^3^	Anti-inflammatory and antibiotic	[26]
*Gracilaria* spp.	*G. birdiae*	Hot water extraction (90 °C)	3.7 × 105 g mol^−1^	-	-	85.6	8.4	65.4% (Gal), 25.1% (3,6-AG) ^4^	Antioxidant	[87]
	*G. caudata*	Enzymatic extraction; Papain (60 °C); SP-Gc	116.51	24.96	1.63	85.6	-	-	In vitro and in vivo antioxidant	[83]
	*G. chouae*	Hot water extraction (90 °C)	715.5	24.2	-	52.63	7.9	Gal: 3,6-AG 1:0.6	Antioxidant	[70]
	*G. cornea*	Enzymatic extraction; Papain (60 °C)	18	-	ND	68.8	15.66	-	Antinociceptive and anti-inflammatory	[84]
	*G. debilis*	Hot water extraction (80 °C)	44	-	14.2	52.65	15.66	-	Antioxidant and anticoagulant	[24]
		DEAE column (1.5 × 50 cm), Fractionated	32	-	-	59.7	14.08	-		
		Sepharose 4LB (1.5 × 50 cm) column, Purification	20	-	23.5	67.6	16.01	-		
	*G. fisheri*	RT water extraction	-	5.41	3.54	66.38	6.86	-	Antioxidant	[77]
		55 °C water extraction	-	9.66	3.06	86.79	8.95	-		
	*G. fisheri*	25 °C water extraction	-	6.03	3.33	67.09	7.05	-	Antioxidant	[31]
		55 °C water extraction	-	9.49	2.73	85.99	10.71	-		
		DEAE-Toyopearl 650 M column: F1	-	-	2.79	56.33	16.39	-		
		DEAE-Toyopearl 650 M column: F2	-	-	2.62	56.63	20.3	-		
	*G. gracilis*	Hot water extraction (90–95 °C): PGCL	-	-	1.49	32.52	-	3.33% (Gal), 6.16% (Glu), 0.06% (Xyl), 6.89% (Ara) ^5^, 7.03% (Rib) ^6^	Antioxidant	[67]
	*G. lemaneiformis*	Hot citric acid extraction (100 °C)	-	22.55	-	-		45.84% (Gal), 35.17% (Glu), 18.12% (Xyl), 0.33% (Man) ^7^	Immuno-modulatory	[47]
		DEAE-Sepharose fast-flow ion exchange column; GLP-2 (0.4 NaCl elution)	-	17.5	1.47	82.4	3.06	-		
	*G. lemaneiformis*	Autoclaved under high pressure (121 °C); GLSP	71.87	2.77	0.39	98.85	11.26	-	Anti-diarrhea on ETEC-K88 infected mice	[52]
	*G. lemaneiformis*	Hot citric acid extraction (100 °C)	-	-	1.72	59.93	29.82	Gal: Glu: Man: Xyl: Fuc ^8^: Ara; 71.78:13.48:4.23:7.92: 1.45:1.14	In vitro digestibility and prebiotic activities	[49]
	*G. lemaneiformis*	Hot citric acid extraction (100 °C)	31.5	-	1.6	59	23	Gal: Glu: Xyl; 21.1:4.76:1,	Regulating lipid metabolism	[34]
	*G. lemaneiformis*	Ultrasonic (50 W, 40 KHz)-microwave (800 W, 2450 MHz)-assisted extraction	-	34.8	-	-	-	Mainly composed of galactose	Anti-tumor	[65]
		DEAE Sephadex A-50; GLP-1	5.5	14.3	-	-	0	92.5% (Gal), 7.6% (Ara)		
		DEAE Sephadex A-50; GLP-2	85	65.2	-	-	10.8	82.8% (Gal), 13.4% (Glu), 3.8% (Ara)		
	*G. ornata (Go)*	Go-1 (25 °C)	-	9.2	3.7	33.14	58.8	-	Anti-microbial	[88]
		Go-2 (80 °C)	-	4.1	1.3	57.71	10.3	-		
		Go-3 (80 °C)	-	4.9	0.1	62.2	10.3	-		
	*G. rubra*	Hot water extraction (90 °C)	-	-	0.46	47.16	12.42	-	Antioxidant and immuno-modulatory	[30]
		DEAE-52 cellulose and Sephadex G-50: GRPS-1-1	1310	-	0.16	45.61	5.96	1.79:1 (Gal: Fuc)		
		DEAE-52 cellulose and Sephadex G-50: GRPS-2-1	691	-	ND	54.78	8.46	2.16:1 (Gal: Fuc)		
		DEAE-52 cellulose and Sephadex G-50: GRPS-3-2	923	-	ND	52.72	12.03	2.76:1 (Gal: Fuc)		
	*G. verrucose*	Hot water extraction (55 °C); GSP	≤20.0	7.6	-	83.8	13.1	93.5% (Gal), 2.2% (Glu), 2.7% (Man), 13.4% (3,6-AG)	Inhibitory effect on the growth and adhesion of *Escherichia coli* (ETEC) K88	[53]
*Porphyra* spp.	*P. haitanensis*	Autoclaved under high pressure (121 °C); PHSP	165.45	4.11	0.64	99.12	10.94	-	Anti-diarrhea on ETEC-K88 infected mice	[52]
	*P. haitanensis*	Microwave-assisted extraction (77.84W)	-	5.01	ND	75.36	-	Gal: Glu: Man: Xyl: Ara: Rha ^9^; 11.55:9.9:12.45:1:9.38:10.25	In vivo (SGC-7901 cells) and in vitro antitumor	[80]
	*P. haitanensis*	Hot water extraction	-	-	-	78.9	17.7	10.5 (3,6-AG)	Antioxidant	[89]
		Acetylation	-	85.7	-	74.4	14.1	9.7 (3,6-AG)		
		Phosphorylation	-	49.8	-	71.2	11.6	8.9 (3,6-AG)		
		Benzoylation	-	39.7	-	68.9	9.3	6.3 (3,6-AG)		
	*P. haitanensis*	Hot water extraction, fractionated with alkaline modification; F1	850	-	-	60.4	17.4	Gal: Fuc 92:8 3,6-AG 12.4%	In vivo antioxidant	[90]
	*P. haitanensis*	DEAE-cellulose (3.5 × 50 cm); F1 (0.1 M Nacl elution)	-	44.6	-	60.4	17.4	12.4% *w*/*w* (3,6-AG)	Antioxidant	[91]
		DEAE-cellulose (3.5 × 50 cm); F2 (0.5 M Nacl elution)	-	47.8	-	50.6	20.5	4.7% *w*/*w* (3,6-AG)		
		DEAE-cellulose (3.5 × 50 cm); F3 (1.0 M Nacl elution)	-	7.5	-	41.2	33.5	1.5% *w*/*w* (3,6-AG)		
	*P. haitanensis*	Hot water extraction (80 °C)	630	-	0.056 g/L	85	2.7 mg/mL	Gal: Glu: Fuc; 76.2:2.1:1	Antioxidant	[75]
	*P. yezoensis*	Ultrasonic pretreatment (20 kHz, 400–1200 W, 2–8 s/2 s (on/off) w) and hot water extraction	Near 300	6.24	-	94.5	7.2	Gal: Xyl: Fuc: Ara; 46: 2:1:1, 3,6-AG 12.3%	Antiproliferation of cancer cell lines (SGC-7901, 95D)	[64]
	*P. yezoensis*	Hot water extraction (95 °C): Normal	224	10.6	-	-	10.7	Gal: Glu: Xyl: Fuc; 95.1:2.2: ND: 1.5, 14.2 (3,6-AG)	Antioxidant and anti-inflammatory	[25]
		Hot water extraction (95 °C): Discolored nori	30	20.6	-	-	10.5	Gal: Glu: Xyl: Fuc; 86.4: ND: ND: 7.6, 24.7 (3,6-AG)		
*Ahnfeltia* spp.	*A. plicata*	Low-alkaline extraction (NaBH_4_)	550	16.0	-	96.5	35.5	48.30% (3,6-AG)	Antioxidant and in vitro anticoagulant	[16]
	*A. plicata*	Pretreated by ultrasound in the Citric acid–sodium citrate buffer solution (agar)	-	24.53	-	-	0.55	40.08% (3,6-AG)	-	[86]
		Polyethylene glycol (PEG) precipitation; PA (agarose)	-	13.86	-	-	0.28	51.85% (3,6-AG)	-	
		DEAE-cellulose (3.5 × 50 cm); DA (agarose)	-	14.48	-	-	0.15	53.49% (3,6-AG)	-	
		PEG combined with DA; PDA (agarose)	-	10.44	-	-	0.07	57.78% (3,6-AG)	-	
	*A. tobuchiensis*	Alkaline treatment with NaBH_4_				96	0.6	58% Galactose, 38% (3,6-AG)	-	[92]

^1^ Galactose. ^2^ Xylose. ^3^ Glucose. ^4^ 3,6-anhydro-α-l-galactose. ^5^ Arabinose. ^6^ Ribose. ^7^ Mannose. ^8^ Fucose. ^9^ Rhamnose.

**Table 2 biomolecules-13-01745-t002:** Summary of anti-cancer and anti-tumor effects of SGs from seaweed of agarophytes.

Seaweed Species	Main Compounds	Model Method (Cell Lines)	Proposed Mechanism of Action	Reference
*Gracilaria edulis*	Sulfated polysaccharides	Human lung adenocarcinoma (A549), human hepatic carcinoma (HepG2), and prostate cancer (PC3)	Decreased % cell viability, increased % LDH ^1^ release, shrunk cell morphology and echinoid-shaped cell structure, and changed histopathological in rat organs	[19]
*Gracilaria fisheri*	Sulfated polysaccharides	Cholangiocarcinoma (CCA) cells; (HuCCA-1, RMCCA-1, and KKU-M213)	Inhibited CAA cell proliferation, inhibited EGFR ^2^ and ERK ^3^ phosphorylation, retarded the cell at G0/G1 phase, decreased protein levels of cyclin-D, cyclin-E, increased expression level of the tumor suppressor protein P53 and cyclin-dependent kinase inhibitor P21	[20]
*Gracilaria fisheri*	Sulfated polysaccharides	Cholangiocarcinoma (CCA) cells; (HuCCA-1, RMCCA-1, and KKU-M213)	Reduced migration rate of CCA, induced phosphorylation of EGFR, ERK, and FAK ^4^	[21]
*Gracilaria foliifera*	Sulfated polysaccharides	Three human breast cancer (MDA-MB-231, MCF-7, and T-47D)	Increased % of cytotoxicity, and inhibited the cell growth	[110]
*Gracilaria lemaneiformis*	Sulfated polysaccharides	Human breast cancer (MCF-7), human hepatic carcinoma, cervical cancer Hela, and Madin–Darby canine kidney	Increased inhibitory ratio	[63]
*Porphyra haitanensis*	Sulfated polysaccharides	Six-day-old S180 ascites tumor cells implanted ICR mice	Inhibited the growth of transplanted tumor, increased the weight of spleen, increased level of TNF-α ^5^ and NO ^6^, and increased the antitumor activity	[105]
*Porphyra yezoensis*	Sulfated polysaccharides	Human gastric cancer (SCG7901) and human lung cancer (95D)	Degradation of polysaccharides affected the activity and decreased % cell viability	[62]

^1^ Lactate dehydrogenase. ^2^ Epidermal growth factor receptors. ^3^ Extracellular signal-regulated kinase. ^4^ Fructokinase enzyme. ^5^ Tumor necrosis factor- α. ^6^ Nitric oxides.

**Table 3 biomolecules-13-01745-t003:** Summary of anticoagulant effects of SGs from seaweed of agarophytes.

Seaweed Species	Main Compounds	Model Method	Proposed Mechanism of Action	Reference
*Gelidium crinale*	Sulfated polysaccharides	Normal human plasma	Decreased APTT ^1^ and inactivated thrombin or factor X_a_	[15]
*Gracilaria birdiae*	Sulfated polysaccharides	Normal citrate-treated human plasma, activated partial thromboplastin time (APTT), and prothrombin time (PT)	High molecular weight (>30 kDa) is more affected, and decreased APTT, TT ^2^, and Heptest	[23]
*Gracilaria debilis*	Sulfated polysaccharides	Normal human plasma	Prolonged APTT and no inhibition for PT ^3^	[24]
*Porphyra haitanensis*	Sulfated polysaccharides	Citrate normal chicken plasma	Prolonged APTT, TT, and PT	[89]
*Porphyra perforata*	Sulfated polysaccharides	New Zealand rabbit blood	Increased APTT index and PT, and shortened PT value	[113]
*Ahnfeltia plicata*	Agarose with sulfation	Normal citrate-treated human plasma and activated partial thromboplastin time (APTT)	Prolonged APTT value close to heparin	[16]

^1^ Activated partial thromboplastin time. ^2^ Thrombin time. ^3^ Prothrombin time.

**Table 5 biomolecules-13-01745-t005:** Summary of anti-lipidermic, anti-obesity, and anti-diabetic effects of SGs from seaweed of agarophytes.

Seaweed Species	Main Compounds	Model Method	Proposed Mechanism of Action	Reference
*Gracilaria lemaneiformis*	Sulfated polysaccharides	High-fat-diet (HDF)-induced mice	Decrease serum TC ^1^, TG ^2^, and free fatty acid levels; lowered serum alanine aminotransferase and aspartate aminotransferase; controlled body weight and alleviated liver injury by decreasing serum alanine upregulating the expression of the PPARα ^3^, ACS ^4^, and CPT1a ^5^ gene	[34]
*Gracilaria lemaneiformis*	Sulfated polysaccharides	High-fat-diet (HDF)-induced mice	Reduced weight gain, improved body composition by increased colon length, reduced the level of TC and LDL-C ^6^, increased the level of HDL-C ^7^, reduced fat accumulation in the liver and adipose tissue, repaired damaged intestine from HFD-induced, enhanced positive gut microbiota and restored HFD-induced dybiosis of the gut microbiota, promoted utilization of polysaccharide and lipid secretion, and improved lipid metabolism	[38]
*Gracilaria opuntial*	Sulfated polygalactans	In vitro antidiabetic activities	Inhibited α-amylase, α-glucosidase, and dipeptidyl peptidase-4, and suppressed the hyperglycemic response in diabetic condition	[41]
*Porphyra haitanensis*	Porphyran	Alloxan-induced diabetic mice	Decreased blood glucose and body weight; reduced the levels of TC and LDL; increased the level of HDL; alleviated the damage of tissue in diabetic mice; decreased the MDA ^8^ level in the serum, liver, and kidney; promoted antioxidant activity and lipid metabolism; and stimulated β-cell proliferation	[35]
*Porphyra haitanensis*	Porphyran	High-fat-diet (HDF)-induced mice	Reduced plasma HDL-C; decreased the level of MDA in liver; increased the activities of superoxide dismutase, catalase, and glutathione peroxidase; and increased the levels of serum SOD ^9^ and GSH-Px ^10^	[37]
*Porphyra yezoensis*	Porphyran	High-fat-diet (HDF)-induced ICR mice	Decreased the percentage of body weight gain and serum lipid profile, increased fecal cholesterol and TG excretion, and alleviated liver damage induced by the diet	[36]
*Porphyra yezoensis*	Porphyran	High-fat-diet (HDF)-induced diabetic KK-Ay mice	Reduced spleen weight, increase cecum weight, decreased plasma insulin levels and the calculated HOMA-IR ^11^, increased plasma adiponectin level, improved glucose metabolism and insulin resistant, increased SCFAs ^12^, and enhanced *Bacteroides*, but reduced *Clostridium coccoides*	[17]

^1^ Total cholesterol. ^2^ Triglyceride. ^3^ Peroxisome proliferator-activated receptor alpha. ^4^ Acute coronary syndrome. ^5^ Carnitine palmitoyltransferase Iα. ^6^ Low-density lipoprotein cholesterol. ^7^ High-density lipoprotein cholesterol. ^8^ Malondialdehyde. ^9^ Superoxide dismutase. ^10^ Glutathione peroxidase. ^11^ Homeostatic model assessment of insulin resistance. ^12^ Short-chain fatty acid.

**Table 6 biomolecules-13-01745-t006:** Summary of anti-microbial effects of SGs from seaweed of agarophytes on bacteria strains.

Seaweed Species	Main Compounds	Proposed Mechanism of Action	Reference
*Gracilaria corticata*	Sulfated polysaccharides	Gamma irradiant White spot syndrome virus (WSSV) infection in juvenile *Litopenaeus vannamei*	[44]
*Gracilaria fisheri*	Sulfated galactans	White spot syndrome virus (WSSV) infection in shrimp (*Penaeus monodon*) haemocytes	[42]
*Gracilaria fisheri*	Sulfated galactans supplement diet	Enhanced shrimp *Penaeus vanamei* immunity by inhibited *Vibrio parahaemolyticus* infection	[43]
*Gracilaria ornata*	Crude sulfated polysaccharides	*Escherichia coli*	[88]
*Gracilaria verrucosa*	Sulfated galactans	Inhibit the adhesion of Diarrheagenic enterotoxigenic *Escherichia coli* (ETEC) K88 on yeast cells, promoted nucleic acid release and the fluorescence quenching of membrane proteins, and preventing the invasion of diarrheagenic bacteria on intestinal epithelium	[52]
*Gracilaria verrucosa*	Sulfated galactans	Enterotoxigenic *Escherichia coli* (ETEC) K88	[54]

**Table 7 biomolecules-13-01745-t007:** Summary of immunostimulating activities of SGs from seaweed of agarophytes..

Seaweed Species	Main Compounds	Model Method	Proposed Mechanism of Action	Reference
*Gracilaria birdiae*	Sulfated polysaccharides	White spot syndrome virus (WSSV) infected shrimp (*Litopenaeus vannamei*)	Promoted a delayed immunostimulation, enhanced the agglutinating capacity and PO ^1^ activity, and increased the survival rate	[122]
*Gracilaria fisheri*	Sulfated galactan	WSSV infected shrimp (*Penaeus monodon*)	Enhanced immune parameters; increased the level of THC ^2^, PO, SOD ^3^, and O^2−^ activities; enhanced total haemocytes, phenoloxidase activity, superoxide anions, and superoxide dismutase; and exhibited immune stimulatory	[45]
*Gracilaria lemaneiformis*	Sulfated polysaccharides	RAW 264.7 macrophage cell	Improved proliferation and pinocytic capability of cell and promoted the production of ROS ^4^, NO ^5^, IL-6 ^6^, TNF-α ^7^	[47]
*Gracilaria lemaneiformis*	Sulfated polysaccharides	Rat mast cell line (RBL-2H3), human basophil cell line (KU812 cells), and TM-sensitized BALB/c mice	Alleviated allergy symptoms, reduced TM-specific IgE ^8^ and IgG1 ^9^, suppressed Th2 cell ^10^ polarization, promoted the function of Treg cell, inhibited the function of RBL-2H3 cells ^11^, suppressed p38 MAPK ^12^, inhibited the secretion of IL-4 ^13^ and TNF-α, decreased the score of the anaphylactic response and diarrhea rates, and diminished the concentration of histamine and mMCP-1 ^14^	[123]
*Gracilaria lemaneiformis*	Sulfated polysaccharides	Anti-allergy Ovalbumin (OVA)-induced mouse model	Alleviated intestinal villi injury, reduced OVA-specific immunoglobulin E, histamine, and mast cell protease-1 levels in the serum, reduced the level of serum interleukin-4, increased serum interferon-γ level, decreased B cell and mast cell populations, and upregulated the regulatory T cells	[124]
*Gracilaria rubra*	Sulfated polysaccharides	LPS-induced RAW 264.7 macrophage cell	Enhanced phagocytic activity, acid phosphatase activity, and NO production; strong immunostimulating effects (high sulfate content)	[30]
*Porphyra haitanensis*	Sulfated polysaccharides	TM-induced mouse allergy model	Suppressed allergic reaction; modulated the serum levels of IgE, IgG1, and IgG2 ^15^; decreased histamine release; repaired the pathology in the jejunum; reduced Th2 cytokines; modulated Th1/Th2 ^16^ balance; and induced the secretion of IFN- γ ^17^	[46]
*Porphyra vietnamensis*	Sulfated polysaccharides	Wistar albino rats and albino mice	Increased the weight of thymus, spleen, and lymphoid organ cellularity, increased total leucocyte and lymphocyte count, enhanced antibody titer values, and decreased response to DTH ^18^ reaction	[18]
*Porphyra haitanensis*	Sulfated polysaccharides	RAW 264.7 macrophage cell and BALB/c murine model	Increased the phagocytosis activity; enhanced the secretion of IL-6, IL-10, and TNF-α; reduced NO secretion; decreased CD8+ T cells ^19^; enhanced the subpopulation of CD4+ T cells ^20^, DCs ^21^ and Tregs ^22^; promoted the proliferation of mice lymphocytes; and induced the generation of TNF-α and IL-10 in vivo	[66]

^1^ Phenoloxidase. ^2^ Total haemocyte count. ^3^ Superoxide dismutase. ^4^ Reactive oxygen species. ^5^ Nitric oxide. ^6^ Interleukin-6. ^7^ Tumor necrosis factor- α. ^8^ Immunoglobin E. ^9^ Immunoglobin G. ^10^ T-helper2 cells. ^11^ β-hexosaminidase in RBL-2H3 cells. ^12^ Mitogen-activated protein kinase. ^13^ Interleukin-4. ^14^ Monocyte chemoattractant protein 1. ^15^ Immunoglobulin G subclass level 2. ^16^ Predominant T cell subsets. ^17^ Interferon gamma. ^18^ CD8+T lymphocytes. ^19^ CD4+T lymphocytes. ^20^ Dendritic cells. ^21^ Regulatory T cells. ^22^ Delayed-Type Hypersensitivity.

**Table 8 biomolecules-13-01745-t008:** Summary of the influence on the gut microbiota properties of SGs from seaweed of agarophytes.

Seaweed Species	Main Compounds	Model Method	Proposed Mechanism of Action	Reference
*Gelidium pacificum Okamura*	Sulfated polysaccharides	Mice with antibiotic-associated diarrhea (AAD)	Increased the richness and diversity of the gut microbiome; altered the composition of the gut microbiota; increased the relative abundance of Bacteroides, *Oscillospira*, *Bifidobacterium*; decreased the relative abundance of *Parabacteroides*, *Sutterella*, AF12; improved metabolic pathway of gut microbiota; down-regulated the concentrations of inflammatory cytokines (TNF-α ^1^, IL-1β ^2^, IL-2 ^3^); and improved contents of SCFAs ^4^	[48]
*Gracilaria lemaneiformis*	Sulfated polysaccharides	In vitro stimulated digestion and in vitro fermentation by human fecal microbiota	Modulated the composition of gut microbes; might change the metagenomic function characteristics of microbes and not have significant changes when passing through the simulated digestion system	[49]
*Gracilaria rubra*	Sulfated polysaccharides	In vitro stimulated digestion and in vitro fermentation by human fecal microbiota	Decreased pH level, increased SCFA concentration, modulated the structure of gut microbiota, and increased the relative abundances of bacteria	[126]

^1^ Tumor necrosis factor- α. ^2^ Interleukin-1 beta. ^3^ Interleukin-2. ^4^ Short-chain fatty acids.

**Table 9 biomolecules-13-01745-t009:** Summary of anti-diarrhea activities of SGs from seaweed of agarophytes.

Seaweed Species	Main Compounds	Model Method	Proposed Mechanism of Action	Reference
*Gracilaria birdiae*	Sulfated polysaccharides	Naproxen-induced gastrointestinal damage in Wistar rats	Reduced the macroscopic and microscopic naproxen-induced gastrointestinal damage, increased gastric mucosal resistance, decreased aggressive factors, increased endogenous GSH levels ^1^, and inhibited inflammatory cell infiltration and lipid oxidation	[85]
*Gracilaria cervicornis*	Sulfated polysaccharides	Trinitrobenzenesulfonic acid (TNBS)-induced colitis in rats	Reduced the macroscopic and microscopic TNBS-induced intestinal damage, avoided the consumption of GSH, decreased pro-inflammatory cytokine levels, MDA ^2^ and NO_3_/NO_2_ ^3^ concentrations, and diminished the MPO ^4^ activity	[51]
*Gracilaria caudata*	Sulfated polysaccharides	Acute diarrhea was induced in Wistar rats by administration of castor oil, and Swiss mice induced with activated charcoal	Reduced fecal mass, diarrheal feces, and enteropooling; reduced cholinergic mechanism, and interacted with both the GM1 receptor ^5^ and CT ^6^	[50]
*Gracilaria lemaneiformis*, *Porphyra haitanensis*	Sulfated polysaccharides	ETEC-K88 (*E. coli*)-infected BALB/c mice	Alleviated mice diarrhea symptoms, inhibited the release of pro-inflammatory cytokines (TNF-a ^7^, IL-6 ^8^), suppressed the secretion of immunoglobulin A, and decreased nitroblue tetrazolium levels	[52]

^1^ Glutathione levels. ^2^ Malonyl dialdehyde. ^3^ Nitrate and nitrite. ^4^ Myeloperoxidase. ^5^ Monosialotetrahexosyl ganglioside receptor. ^6^ Interleukin-1 beta. ^7^ Cholera toxin. ^8^ Interleukin-6.

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
