# Peer review of "Sulfated Galactans from Agarophytes: Review of Extraction Methods, Structural Features, and Biological Activities"

_biomolecules, 2023, doi:10.3390/biom13121745_

Round 1

Reviewer 1 Report

Comments and Suggestions for Authors

The authors describe the physiological activity, extraction method, and structural characteristics of sulfated galactans from agarophytes.

In particular, the part that describes the physiological activity and structural characteristics according to the extraction method is thought to have well organized and described the information necessary to secure industrially useful ingredients.

However, I hope that the principle of physiological activity shown by the described material is expressed in pictures to make it easier to distinguish.

Author Response

Dear reviewer, thanks for your keen review. We revised the manuscript according to your comments and highlighted in Green color. Figure 7 was added to the manuscript to display the mechanism of physiological activities.

Reviewer 2 Report

Comments and Suggestions for Authors

The review concerns sulfated galactans from agarophytes and covers extraction methods, structure characteristics and their biological properties. Sulfated galactans, charged with numerous sulfate groups, revealed anti-tumor, anti-coagulant, anti-inflammatory, antioxidant, anti-obesity, anti-diabetic, anti-microbial, anti-diarrhea and other sorts of activities. The author note that chemical and biological properties of sulfated galactanes depend on taxonomy and ecological factors including environmental factors and harvest period, as well as isolation methods including pretreatment (delipidisation, deproteination, depigmentation), extraction, and conditions of separation. The paper collects and systematically reviews the scientific data about sulfated galactans, that may be useful for further industrial application of agarophytes not only as the source of agar production, but also for functional food and pharmacy. The review is very interesting, good illustrated by tables and figures, English is excellent but several imperfections should be fixed.

1) Line 17. Replace “Meanwhile, ecological factors, i.e., the taxonomy, environmental factors, and…” with “Meanwhile, the taxonomy, ecological factors, i.e. environmental factors, and…”. Remember, please, forever that taxonomy is not an ecology and it derives from the genetics!

2) Line 271. Porphyra spp. should be replaced with Porphyra spp. i.e. italized, please, the genus name.

3) Line 272. The same error.

4) Line 377. Replace, please, 5-Fluoracila with “5-fluoracil”.

5) Line 404. Replace, please, SO3 with SO3, i.e. use, please, subscript style for numbersin chemical formulae. Check, please, the entire text for similar errors incliding the tables.Replace

6) Line 406. Replace NaBH4 with NaBH4, i.e. use the subscript.

7) Line 420. Replace NO2/NO3 with NO2/NO3, i.e. use, please, the subscript.

8) Line 620. Replace “First, the influence of ecological factors of the taxonomy, growth environmental factors and harvest period” with “First, the influence of taxonomy and ecological factors including growth environmental factors and harvest period”. Note, please, again that taxonomy is not an ecology and it derives from the genetics. It is hard general biology error!

9) Lines 632–634. This text should be deleted as senseless.

10) Check and correct each reference in the list of the references. Just open any fresh article published in the Biomolecules and carefully compare your references with the sample. Use, please, correctly commas, points, letter size, long dashes between the page numbers instead of short etc. because each reference is erroneous. Be more focused, please.

 The article is very interesting, comprehensive and may be published after minor revise.

Author Response

Dear reviewer, thanks for your keen review. We revised the manuscript according to your comments and highlighted in Yellow color. The point-by-point responses are as follows.

Q1: Line 17. Replace “Meanwhile, ecological factors, i.e., the taxonomy, environmental factors, and…” with “Meanwhile, the taxonomy, ecological factors, i.e. environmental factors, and…”. Remember, please, forever that taxonomy is not an ecology and it derives from the genetics!

A1: Thanks for your correction. We revised them in Line 16-17.

Q2: Line 271. Porphyra spp. should be replaced with Porphyra spp. i.e. italized, please, the genus name.

A2: We revised them in Line 269.

Q3: Line 272. The same error.

A3: We revised them in Line 270.

Q4: Line 377. Replace, please, 5-Fluoracila with “5-fluoracil”.

A4: Thanks for your correction. We revised them in Line 375.

Q5: Line 404. Replace, please, SO3 with SO3, i.e. use, please, subscript style for numbersin chemical formulae. Check, please, the entire text for similar errors incliding the tables. Replace

A5: We revised them in Line 402 and checked similar errors including table 4.

Q6: Line 406. Replace NaBH4 with NaBH4, i.e. use the subscript.

A6: We revised them in Line 404.

Q7: Line 420. Replace NO2/NO3 with NO2/NO3, i.e. use, please, the subscript.

A7: We revised them in Line 425 and table 4.

Q8: Line 620. Replace “First, the influence of ecological factors of the taxonomy, growth environmental factors and harvest period” with “First, the influence of taxonomy and ecological factors including growth environmental factors and harvest period”. Note, please, again that taxonomy is not an ecology and it derives from the genetics. It is hard general biology error!

A8: Thanks for your correction. We revised them in Line 625-626.

Q9: Lines 632–634. This text should be deleted as senseless.

A9: We delete them already.

Q10: Check and correct each reference in the list of the references. Just open any fresh article published in the Biomolecules and carefully compare your references with the sample. Use, please, correctly commas, points, letter size, long dashes between the page numbers instead of short etc. because each reference is erroneous. Be more focused, please.

A10: Thanks for your correction. We correct each reference follow the pattern you suggest.